# Guidelines for Pregnancy Management During the COVID-19 Pandemic: A Public Health Conundrum

**DOI:** 10.3390/ijerph17218277

**Published:** 2020-11-09

**Authors:** Caroline Benski, Daria Di Filippo, Gianmarco Taraschi, Michael R. Reich

**Affiliations:** 1Takemi Program in International Health, Harvard T.H. Chan School of Public Health, Boston, MA 02115, USA; reich@hsph.harvard.edu; 2Department of Obstetrics and Gynecology, Geneva University Hospital, 1205 Geneva, Switzerland; g.m.t.89@hotmail.it; 3School of Women’s and Children’s Health, University of New South Wales, Sydney, NSW 2031, Australia; dariadifilippodaria@gmail.com

**Keywords:** antenatal care, intrapartum care, postnatal care, COVID-19, Sars-CoV-2, pandemic, obstetrics guidelines, public health

## Abstract

Pregnant women seem to be at risk for developing complications from COVID-19. Given the limited knowledge about the impact of COVID-19 on pregnancy, management guidelines are fundamental. Our aim was to examine the obstetrics guidelines released from December 2019 to April 2020 to compare their recommendations and to assess how useful they could be to maternal health workers. We reviewed 11 guidelines on obstetrics management, assessing four domains: (1) timeliness: the time between the declaration of pandemics by WHO and a guideline release and update; (2) accessibility: the readiness to access a guideline by searching it on a common browser; (3) completeness: the amount of foundational topics covered; and (4) consistency: the agreement among different guidelines. In terms of timeliness, the Royal College of Obstetricians and Gynaecologists (RCOG) was the first organization to release their recommendation. Only four guidelines were accessible with one click, while only 6/11 guidelines covered more than 80% of the 30 foundational topics we identified. For consistency, the study highlights the existence of 10 points of conflict among the recommendations. The present research revealed a lack of uniformity and consistency, resulting in potentially challenging decisions for healthcare providers.

## 1. Introduction

COVID-19 is an infectious disease caused by SARS-CoV-2, the seventh coronavirus with proven inter-human transmission [1]. At the end of December 2019, the virus spread rapidly from China to the entire world, with 215 countries affected at the time of this writing (compared to 29 countries affected by the 2003 SARS pandemic). The World Health Organization (WHO) declared COVID-19 a global pandemic on March 11, 2020 [2,3]. Our data collection stopped at the end of April 2020, at which time almost two million people worldwide had a confirmed SARS-CoV2 infection and 130,000 had died of COVID-19. As of August 2020, the number of individuals infected had risen to a total of more than 22 million and 778,000 had died of COVID-19 [4].

Pregnant women have a higher risk of viral infectious diseases because of the physiological changes occurring during pregnancy in the respiratory, reproductive, endocrine, and immune systems [5]. For all coronavirus (CoV) infections, higher rates of negative outcomes have been described for pregnant women [6]. The inflammatory nature of COVID-19 during pregnancy exposes both women and their fetuses to a higher risk of obstetric complications and potentially to long-term multi-systemic complications in exposed newborns. Although several case reports have been published, little is known about the real impact of COVID-19 in pregnancy. No clear evidence for vertical transmission has been reported, and meager data exist regarding the effect of SARS-CoV-2 infection during the first and second trimesters of pregnancy [5,6,7,8,9,10,11]. Even for delivery timing and type (vaginal or caesarean), there is a lack of evidence; therefore, decisions about route of delivery and delivery timing should be individualized for the specific patient based on obstetrical indications and maternal–fetal status.

High rates of caesarean section, performed mainly for fetal distress, have been reported for mothers testing positive for SARS-CoV-2 [12]. Induced preterm delivery, possibly due to COVID-19 associated respiratory failure in late pregnancies, has been described as a reason for concern [13]. 

Until a vaccine and effective therapies are established and widely used, prevention and disease containment measures are the only available strategies. Hence, in the context of the COVID-19 pandemic, healthcare guidelines play a vital role in the management of pregnant patients. Several clinical guidelines for the management of pregnancy during the COVID-19 pandemic have been published since the beginning of the pandemic by multiple international agencies and national authorities. Although policymakers have access to the same references and scientific reports, recommendations on the antenatal, intrapartum, and postnatal/postpartum management vary substantially among these guidelines. 

The aim of this study was to examine the obstetrics guidelines released during the first four months of the COVID-19 pandemic (December 2019 to April 2020) to compare the recommendations and assess their usefulness to maternal healthcare workers. During times of major uncertainty, such as those of a pandemic, healthcare professionals need timely, easy to access, and easy to read guidelines. Our analysis considered the timeliness, accessibility, completeness, and consistency of the guidelines published by the most important international organizations, obstetrics and gynecology (OB-GYN) societies, and health ministries of the countries most severely affected by the pandemic up until the end of April 2020. This is a preliminary expert review, undertaken during an early phase of the pandemic, rather than a full systematic review.

## 2. Materials and Methods 

### 2.1. Team

The team comprised three healthcare providers in maternal and newborn health (CB, DDF, GT), who collaborated together on the International Association of Italian Researchers—COVID-19 Literature in Pills (AIRI-CLIP) project [14], plus one health policy analyst (MRR). AIRI-CLIP was launched by AIRIcerca, an organization that provides a novel communication channel between Italian researchers and society. The AIRI-CLIP initiative consists of scientific articles on COVID-19 summarized in English and Italian [15]. While reviewing the available literature for AIRI-CLIP on the management of pregnancy, we (CB, DDF, GT) found the recommendations to be unclear and decided to focus our attention on a comparison of the main guidelines released globally. The health policy analyst (MRR) joined the project to provide advice on the comparative assessment of the guidelines and participate in writing the paper.

### 2.2. Guidelines 

We reviewed the major guidelines on the management of pregnancy published during the COVID-19 pandemic from its beginning until the end of April 2020. Although many countries released guidelines, we chose to analyze those documents issued by the most affected countries up to the end of April 2020: China, Italy, Spain, the UK, and USA. We therefore included guidelines published by the Italian and Spanish Ministries of Health, the Italian Association of Hospital Gynecologists and Obstetricians (AOGOI) and the Italian Society of Gynecology and Obstetrics (SIGO), the Society for Maternal Fetal Medicine (SMFM), a Chinese expert consensus, and the Royal College of Obstetricians and Gynaecologists (RCOG) [16,17,18,19,20,21]. We also considered publications from international and specialist organizations, such as the International Federation of Gynaecology and Obstetrics (FIGO), the American College of Obstetrics and Gynecology (ACOG), and the International Society of Ultrasound of Obstetrics and Gynecology (ISUOG) [22,23,24]. We included these guidelines because we expected their recommendations to stand as a benchmark for other scientific societies. We also included the guidelines from two international public health organizations, the World Health Organization (WHO) and the Centers for Disease Control and Prevention (CDC USA), because of the significant roles these two organizations play in global health [25,26]. In total, we selected 11 guidelines in different formats (guideline, question-and-answer webpage, commentary, divulgation webpage, interim guidance, poster, and communication). 

### 2.3. Domains

The present study was conceived during the COVID-19 pandemic emergency. Healthcare providers were challenged in continuing to deliver efficient care, while trying to understand the best management for pregnant patients exposed to or infected with SARS-CoV-2. Healthcare professionals needed quick, easily accessible, and up-to-date guidelines on how to best manage their pregnant patients. We defined four domains for reviewing and comparing the selected guidelines: timeliness, accessibility, completeness, and consistency.

Timeliness: Clinicians need ready-to-use and up-to-date guidelines. We acknowledge that releasing a guideline is an ongoing process because of the evolving situation of the pandemic; however, the sooner a guideline is released, the more useful it might be for clinicians. Therefore, we assessed each guideline for its release date and last update until the end of April 2020.

Accessibility: Healthcare guidelines should be easy to obtain to be quickly implemented. Therefore, we evaluated the ease of access by searching the guideline on a common browser. We counted the number of clicks necessary to find the guidelines document on the result page. Ideally, a publication should encompass all of the necessary topics without links to other documents that divide the flow of information. We therefore assessed which guidelines covered all of the basic information in one location. We also considered accessibility in terms of the language of the guideline. 

Completeness: By comparing the guidelines, it became evident that certain topics were recurrent. We outlined key points considered in the peer reviewed literature on COVID-19 and from our clinical experience to identify 30 foundational topics for antenatal, delivery/intrapartum, and postnatal/postpartum care. In terms of antenatal care, we identified 13 foundational topics: personal and social hygiene, mask wearing, symptoms, visits planning, partners/visitors, triage point, infection prevention and control, organization of the place of care, diagnostic and imaging, treatment, fetal monitoring, mental health, and telehealth. The 12 topics of the intrapartum care are: infection prevention and control, delivery ward organization, partner, biohazardous material, transportation, corticosteroids, timing of delivery, mode of delivery, analgesia, management of labor and fetal monitoring, cord clamping, and drugs (MgS04, anticoagulants). Lastly, for postnatal care, we considered five essential topics: mother and child separation and IPC, breastfeeding, breast pump, postpartum visits, and family planning. In our opinion, these topics should be considered by any guideline addressing the management of pregnant women during a pandemic. To determine the scientific basis for the recommendations, we noted the number of references cited. 

Consistency: We compared the guidelines and identified differences in the recommendations endorsed. Thus, we examined the consistency among guidelines as a potential factor impacting clinical activity.

### 2.4. The Process: From Conceptualization to Analysis

For each guideline, we assessed the timeliness (date of first publication and last update), accessibility (number of internet clicks needed to access the document from the original search on common browser), and completeness (topics discussed, type of release, number of references, and number of hyper textual links). Lastly, we evaluated the consistency of the different guidelines for each of the foundational topics. This analysis did not assume a scientific position for any of the specific guidelines, but rather sought to identify similarities and differences across the recommendations given since the beginning of the pandemic on the management of pregnancy with concurrent SARS-CoV-2 infection or COVID-19 disease.

## 3. Results

The results of our assessment of timeliness, accessibility, and completeness among the 11 selected guidelines are described below (see Table 1 for details).

The format used for the selected 11 guidelines varied. The WHO guidelines were available in the form of a question-and-answer webpage. The CDC and ACOG guidelines were on a webpage in dissemination format. All of the others were available as pdf files. The variety of formats used for the guidelines reflects the differences in terms of structure and contents among them. 

### 3.1. Timeliness

RCOG released the first recommendation on March 9, 2020 (two days before the WHO declared the pandemic), followed by ISUOG on March 11, and ACOG on March 13. The CDC, SMFM, and Spanish Ministry of Health all released their guidelines March 17, and the WHO released its Q&A the day after. The Chinese expert consensus was released on the 20th of April, whereas the Italian Ministry of Health published guidelines on March 31, and SIGO/AGOI on April 4. RCOG guidelines were most often updated with seven versions released, as opposed to WHO, whose Q&A were never updated. 

### 3.2. Accessibility

Accessibility was defined as the number of clicks necessary to access the guidelines. Only four guidelines (Chinese expert consensus, WHO, CDC, and ACOG) were accessible with one click from the original search on the common browser. Spanish, RCOG, and ISUOG guidelines needed two clicks, SMFM and FIGO were accessible with three clicks, and SIGO/AOGOI needed up to seven clicks.

As for language used, the Italian and Spanish guidelines were written in the local language, i.e., Italian and Spanish. Chinese guidelines were published in English and Chinese. All of the other guidelines considered in this study were written in English. 

### 3.3. Completeness

The completeness of the publications depended on the type of release (with guidelines in pdf format being the most complete and the question-and-answer format the least detailed) and on the number of references cited (0 for WHO, 79 for FIGO). Furthermore, we pointed out 30 foundational topics which were recurrently addressed by different papers. We deemed a guideline complete if it covered more than 80% of these foundational topics defined as above. However, only 6 out of 11 papers (RCOG, ISUOG, FIGO, SIGO/AOGOI, and Chinese expert consensus) could be defined complete as such. We observed that papers written in the form of a guideline and those with more references were also the most complete ones.

Furthermore, RCOG, ACOG, Spanish Ministry of Health, and the WHO provided additional information in the form of complementary hyper textual links, with a range from 2 (Spanish Ministry of Health) to 60 (RCOG). As observed in Figure 1, the completeness of topics was highly variable among the 11 guidelines. 

For antenatal care, seven guidelines covered more than 10 out of 13 topics (~80%), and RCOG and FIGO covered all of them (100%). For intrapartum care, no guideline covered all 12 topics, with RCOG guidelines covering 92% and WHO guidelines covering only 25% of the 12 topics. For postnatal/postpartum care, FIGO and RCOG covered all five topics (100%), and SIGO, ISUOG, Chinese expert consensus, SMFM, and ACOG covered 80% of the topics. The guidelines from the Spanish and Italian Ministries of Health covered only two of the five topics (40%) for postnatal/postpartum care (see Figure 2).

As highlighted in Table 2, no guideline gave indications about family planning, and the majority did not address the management of biohazardous materials, the use of magnesium sulphate and anticoagulant therapies, and the mode of transport of patients from home to healthcare centers. We also observed that only four guidelines discussed the problem of mental health, as well as the management of labor and fetal monitoring.

### 3.4. Consistency

We observed the following differences in guideline recommendations (see Table 3 for additional detail).

#### 3.4.1. Antenatal Care

*Mask wearing*: While some institutions recommend that everyone wear a mask when visiting hospitals or other high-risk environments, ISUOG, RCOG, FIGO, and the Spanish and Italian Ministries of Health reserve mask wearing to pregnant women who have screened positive for SARS-CoV-2 infection, especially during transport to a secondary center (Italy). FIGO also recommends pregnant women, regardless of their SARS-CoV-2 status, to wear a mask as a general precaution when visiting hospitals. The American–Chinese expert consensus urges pregnant women to comply with their local guidance. The WHO question-and-answer webpage does not provide any recommendation regarding mask wearing. 

*Personal and social hygiene*: General personal hygiene and social distance recommendations are found in each guideline. However, we encountered some slight differences among the guidelines, with some institutions recommending at least two meters of distance (RCOG, FIGO) and others one meter (ISUOG, AOGOI). All others recommended generally avoiding social interactions. 

*Antenatal care visits*: Implementing alternative services for antenatal care, such as telehealth, is a generally endorsed recommendation. For patients who have tested positive for SARS-CoV-2 infection, RCOG and FIGO recommend delaying any routine medical visit until the end of the isolation period. In the ISUOG guidelines, the recommendation is to delay antenatal care visits by 14 days or to when one positive test or two negative tests at one-day intervals are obtained. 

*Partner/companion*: In general, position statements vary, allowing none, one, or the least possible number of people to accompany a pregnant woman to her hospital visits. Even though the presence of a companion is reassuring for a pregnant woman, some guidelines strictly forbid the partner’s presence if symptomatic, even in the labor ward (SMFM). 

#### 3.4.2. Intrapartum Care

*Antenatal corticosteroids*: The majority of guidelines call for caution before the administration of corticosteroids to COVID-19 positive and suspected patients, especially in the context of critically ill patients and only after a multidisciplinary discussion with maternal–fetal experts and infectious disease specialists (ISUOG, SMFM, Spanish Ministry of Health, FIGO). However, SMFM allows the use of corticosteroids when deemed necessary before 34 weeks of gestation, while RCOG and the Chinese expert consensus allow their use as needed for standard care. 

*Respiratory analgesia*: Respiratory analgesia is not recommended by the Spanish Ministry of Health and SMFM for COVID-19 positive and suspected patients because of the risk of aerosol generation, while its use is allowed by RCOG.

*Cord clamping*: Almost all of the examined guidelines recommend against delayed cord clamping, in COVID-19 positive and suspected women, except for RCOG and the Spanish Ministry of Health.

*Skin-to-skin practice*: Only three papers (the Spanish Ministry of Health, WHO, and SMFM) address the topic of skin-to-skin policy. Each one of them gives different advice; SMFM advises against the skin-to-skin practice, WHO endorses it, and the Spanish Ministry of Health allows it under certain conditions. 

#### 3.4.3. Postnatal/Postpartum Care

*Mother/child separation*: The Chinese expert consensus recommends 14 days of separation for COVID-19 suspected/positive patients. More permissively, other guidelines allow rooming in for asymptomatic and mildly ill women, while separation is reserved for those with a severe or critical illness (ISUOG, FIGO, Italian Ministry of Health, Spanish Ministry of Health). RCOG and the WHO encourage early rooming in and skin-to-skin contact. 

*Breastfeeding*: In general, all regulating organizations allow for breastfeeding under specific hygiene conditions for COVID-19 suspected/positive patients, except for the Chinese expert consensus, which recommends against direct breastfeeding.

## 4. Discussion

Our study compared the obstetrics guidelines published by the most important international health institutions, OB-GYN societies, and the Ministries of Health of the most severely affected countries from the beginning of the pandemic to the end of April 2020. Acknowledging the scientific uncertainties of SARS-CoV-2 and the continuously evolving data and studies on which such guidelines were based, we considered their timeliness, accessibility, completeness, and consistency as the most important domains for assessment. 

Even if the first guidelines released may have been not as complete as those following, timeliness did not interfere with completeness, especially when the guidelines were frequently updated. The best example from the analysis is represented by the RCOG guidelines, released first and updated seven times in the observed period. Releasing timely information is indeed fundamental to guide HCP; further details can then be provided once more data are available. Accessibility is another feature of paramount importance for guidelines. To best support healthcare professionals in managing their patients in such an evolving scenario, guidelines should be easily accessible from a website/mailing list and written both in English and in the local language. 

For completeness, we defined 30 foundational topics that we estimated should be covered by guidelines in times of a pandemic. Our study highlighted the existence of 10 points of conflict among the recommendations within those 30 foundational topics. This lack of uniformity represents a challenge for healthcare professionals and likely reflects how different countries reacted to the COVID-19 pandemic. As an example, RCOG guidelines are less strict than those released in China. England initially underestimated the pandemic, whereas China, scarred by its experience of SARS, instituted drastic preventative measures from the beginning, even regarding the management of pregnant women and their newborns. 

The management of pregnancy, delivery, and postnatal/postpartum care during the COVID-19 pandemic has differed vastly around the world. There are many reasons for the differences in guidelines from country to country, which are reflective of the dissimilar contexts in which COVID-19 arose and the different healthcare systems in place to address it. The process for developing guidelines depends highly on the history and structure of the country and its healthcare system. It was a major challenge for developing countries to access, assess, and define the right recommendations for obstetric care. In fact, several aspects of some guidelines on maternal and newborn health were not applicable in low resources countries. One of the authors (CB), for example, worked with the Ministry of Health of Madagascar to review and adapt obstetric guidelines to their specific context. 

To be trustworthy, guidelines during uncertainty should be made using a transparent process, based on the best available evidence, and safeguarded against biases and conflicts of interest [27]. To decrease variations in clinical practice and costly adverse events, guidelines are intended to facilitate the practice of local policymakers and healthcare professionals, thus improving the effectiveness and quality of care [28]. Better global coordination of policymakers would help the process of developing common guidelines that are uniformly applicable in all countries. This could be accomplished through the harmonizing role of a leading international body that could define a template to be used to generate guidelines during a pandemic, including the foundational topics, and could receive and release the guidelines from different authorities, in order to update continuously at a central level the shared guidance available worldwide for managing a pandemic.

To our knowledge, this is the first study to compare the timeliness, accessibility, completeness, and consistency of the obstetrics guidelines released during the COVID-19 pandemic. We believe that three core principles are crucial when writing public health guidelines in times of scientific uncertainty, in order of importance: timely release, easy access, and completeness. Furthermore, especially when facing the uncertainties of a pandemic, one international body would need to coordinate with several local publications to guarantee consistency among the different guidelines. The main limitation of this analysis is that we did not perform a systematic review and we based the assessment of the guidelines’ completeness by establishing 30 foundational topics based on our clinical experience. In addition, we did not evaluate the impact of the different guidelines on healthcare providers or patients. With this article, we are also not proposing scientific answers for the best management of pregnancy, delivery, and postnatal/postpartum care, as these are not our skills, nor are we in the position of generating guidelines. 

## 5. Conclusions

A substantial global effort has been put into limiting the spread of COVID-19, and tremendous work has been done by the main international and national health institutions and OB-GYN societies to produce and update material on the management of pregnant women from the antenatal to postnatal/postpartum periods based on limited and evolving scientific knowledge. Nonetheless, a closer evaluation of the guidance provided to both patients and healthcare providers worldwide has revealed a lack of uniformity and consistency, resulting in challenging decisions for healthcare providers. A more systematic process to deliver guidelines, including the principles and topics highlighted in this study, as well as through a coordinated global response, is critical to facilitate the activity of healthcare providers and guarantee that every mother and every newborn receives the best possible care during a pandemic. We strongly believe that lessons learned from the COVID-19 pandemic should guide the development of global strategies to improve the preparedness of the response to future pandemics.

## Figures and Tables

**Figure 1 ijerph-17-08277-f001:**
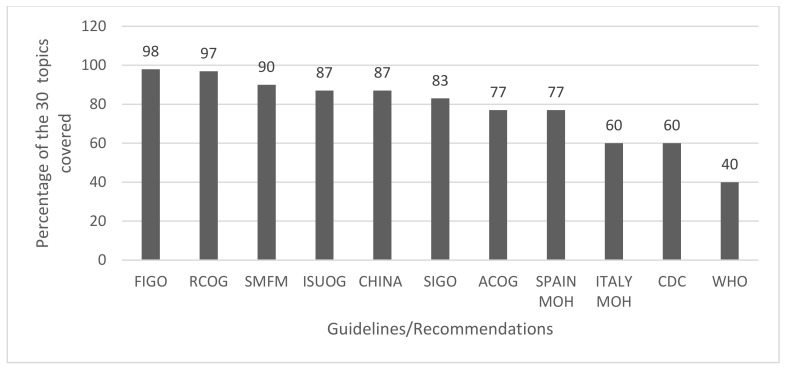
Completeness of foundational topics covered by each recommendation.

**Figure 2 ijerph-17-08277-f002:**
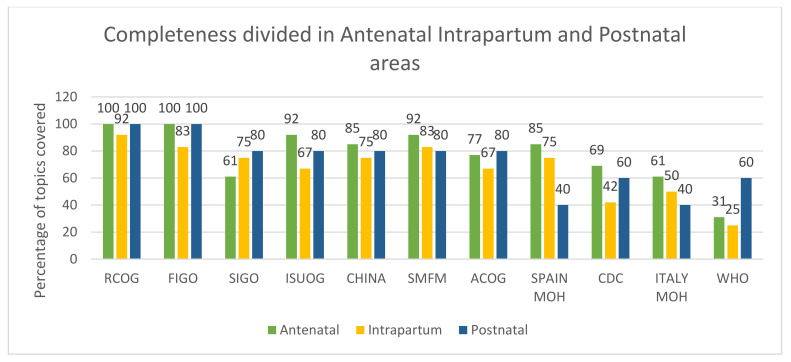
Completeness of topics covered by area.

**Table 1 ijerph-17-08277-t001:** Type of release and recommendations.

Releasing Institution	Country	Release Format	Type of Edition	Publication Title	Publication Date (dd/mm)	Last Update (dd/mm)	Number of Updated Versions	Accessibility (*N* of Clicks)	Number of Topics Covered	Metalinks for Additional Information	No. of References (If Described)
RCOG	UK	WEBPAGE	GUIDELINES	Coronavirus (COVID-19) Infection in Pregnancy	09/03	09/04	7	2	29/30	60	45
ISUOG	N.A.	WEBPAGE	INTERIM GUIDANCE	ISUOG Interim Guidance on 2019 novel coronavirus infection during pregnancy and puerperium: information for healthcare professionals	11/03	-	-	2	26/30	0	64
ACOG	USA	WEBPAGE	PRACTICE ADVICE	Outpatient Assessment and Management for Pregnant Women with Suspected or Confirmed Novel Coronavirus (COVID-19)	13/03	10/04	1	1	23/30	15	1
CDC		WEBPAGE	PRACTICE ADVICE	Considerations for Inpatient Obstetric Healthcare Settings	17/03	06/04	1	1	18/30		76
SMFM		WEBPAGE	COMMENTARY	Coronavirus (COVID-19) and Pregnancy: What Maternal–Fetal Medicine Subspecialists Need to Know	17/03	11/04	1	3	27/30		18
Spanish Ministry of Health	SPAIN	WEBPAGE	GUIDELINES	Management of pregnant women and newborns with COVID-19	17/03	-	-	2	23/30	2	31
WHO	N.A.	WEBPAGE	QUESTIONS AND ANSWERS	Q & A on COVID-19, pregnancy, childbirth and breastfeeding	18/03	-	-	1	12/30	3	0
Chinese Expert Consensus	China	WEBPAGE	EXPERT CONSENSUS	Expert consensus for managing pregnant women and neonates born to mothers with suspected or confirmed novel Coronavirus (COVID-19) infection	20/03	-	-	1	26/30	0	44
Italian Ministry of Health	ITALY	WEBPAGE	COMMUNICATION	COVID-19: guidance for pregnancy, labour, newborns and breastfeeding	31/03	-	-	7	18/30	n.a.	n.a.
SIGO/AOGOI		EMAIL	POSTER	Pregnancy at the time of Coronavirus	04/04			n.a.	25/30	0	n.a.
FIGO	N.A.	WEBPAGE	INTERIM GUIDANCE	Global interim guidance on coronavirus disease 2019 (COVID-19) during pregnancy and puerperium	04/04	-	-	3	29/30	0	72

**Table 2 ijerph-17-08277-t002:** Foundational topics covered by each guideline.

Topics	General Recommendations and Antenatal Care										
	RCOG	ISUOG	ACOG	SMFM	CDC	Spain MoH	WHO	Chinese Expert Consensus	MoH Italy	SIGO/AOGOI	FIGO
Personal hygiene	✓	✓	✓	✓	✓	✓	✓	✓	✓	✓	✓
Social Hygiene (mask)	✓	✓	✓	✓	✓	✓	✓	✓	✓	✓	✓
Symptoms	✓	✓	✓	✓	✓	✓	✓	✓	✓	✓	✓
Organization of the visits	✓	✓	✓	✓	✓	✓	✓	✓	✓	✓	✓
Partner/visitors	✓	✓	✓	✓	✓			✓		✓	✓
Triage point	✓	✓		✓	✓	✓		✓			✓
IPC	✓	✓	✓	✓	✓	✓		✓	✓	✓	✓
Organization of the place of care	✓	✓	✓	✓	✓	✓		✓			✓
Diagnostic (imaging)	✓	✓		✓		✓		✓	✓	✓	✓
Treatment	✓	✓	✓	✓		✓		✓		✓	✓
Fetal monitoring	✓	✓		✓	✓	✓		✓			✓
Mental health	✓	✓	✓								✓
Telehealth	✓		✓	✓		✓					✓
	Intrapartum Care										
IPC	✓	✓	✓	✓	✓	✓	✓	✓	✓	✓	✓
Delivery ward organization	✓	✓		✓	✓	✓		✓	✓	✓	✓
Partner	✓		✓	✓				✓		✓	✓
Biohazardous material	✓							✓		✓	✓
Transportation	✓					✓					
Corticosteroids	✓	✓	✓	✓		✓		✓		✓	✓
Timing of delivery	✓	✓	✓	✓	✓	✓	✓	✓	✓	✓	✓
Mode of delivery	✓	✓	✓	✓	✓	✓	✓	✓	✓	✓	✓
Analgesia	✓	✓	✓	✓	✓	✓		✓	✓	✓	✓
Management of labor and fetal monitoring	✓	✓		✓							✓
Cord clamping	✓	✓	✓	✓		✓		✓	✓	✓	✓
MgS04, anticoagulants	✓		✓	✓							
	Postnatal Care										
Mother/child separation	✓	✓	✓	✓	✓	✓	✓	✓	✓	✓	✓
Breastfeeding	✓	✓	✓	✓	✓	✓	✓	✓	✓	✓	✓
Breast pump	✓	✓	✓	✓	✓			✓		✓	✓
IPC	✓	✓	✓	✓	✓	✓	✓	✓	✓	✓	✓
Postpartum visits	✓	✓	✓	✓			✓	✓		✓	✓
Family planning											

**Table 3 ijerph-17-08277-t003:** Main differences among the 11 guidelines.

**Topics**	**RCOG**	**ISUOG**	**CHINA**	**SPAIN MOH**	**AOGOI/SIGO/Italian MOH**	**ACOG/CDC/SMFM**	**FIGO**	**WHO**
ANTENATAL CARE								
**Mask Wearing**	Pregnant women screened positive for SARS-CoV-2 infection	Apply local guidance	Apply local guidance	Women with acute respiratory symptoms	Pregnant women screened positive for SARS-CoV-2 infection	Women who test positive/PUI women at all times as clinically able *	People visiting a hospital or other high-risk area	NR
**Personal/social Hygiene**	Especially >28 weeks of gestation, distance of at least two meters between individuals	Distance of at least one meter between individuals	Reduce social interactions	Reduce social interactions	Distance of at least one meter between individuals	Reduce social interactions	Distance of at least two meters or six feet between individuals	Reduce social interactions
**Antenatal visits**	Delay visits until the end of self-confinement period	Postpone routine follow-up appointments by 14 days or until positive/two consecutive negative test results	NR	NR	When possible postpone visits, by taking note into the clinical file.	NR	Offer all visits for obstetric emergencies **/postpone routine follow-up appointments by 14 days or until positive/two consecutive negative test results	Guarantee to all pregnant women, including those with confirmed/suspected COVID-19, high quality of care before, during, and after childbirth (including mental health care)
**Partner/companion**	Come alone to the visits or with one person maximum	Consider reducing the number of visitors to the department	Consider reducing the number of visitors to the department.	Come alone to the visits or with one person maximum	Positive partner: notify your obstetric team: access forbidden	Consider reducing the number of visitors to the department ***	Come alone to the visits	NR
	**RCOG**	**ISUOG**	**CHINA**	**SPAIN MOH**	**AOGOI/SIGO/Italian MOH**	**ACOG/CDC/SMFM**	**FIGO**	**WHO**
INTRAPARTUM CARE								
**Antenatal Corticosteroids**	- Steroids for standard care - Do not delay urgent intervention to allow steroid administration	- Avoid steroids in critical patients - Discuss steroids administration with MDC - Avoid tocolysis	Steroids to all viable premature fetuses	- Steroids administration on individual basis - Discuss steroids administration with MTD	NR	- Steroids administration on individual basis - Particular caution for critically ill women in ICU setting	- Avoid steroids in critical patients - Discuss steroids administration with MDC - Avoid tocolysis	NR
**Respiratory Analgesia**	Entonox allowed (with disposable filter)	NR	NR	Respiratory analgesia is not recommended	NR	Consider suspending use of nitrous oxide	NR	NR
**Cord Clamping**	Delayed cord clamping recommended	Prompt cord clamping	Delayed cord-clamping not recommended for women infected with COVID-19	Delayed clamping allowed if maternal and newborn isolation can be done properly	Prompt cord clamping	NR	Prompt cord clamping	NR
**Skin to skin practice**	NR	NR	NR	Skin to skin is allowed if maternal and newborn isolation can be done properly	Avoid skin to skin	NR	NR	Skin to skin is allowed
	**RCOG**	**ISUOG**	**CHINA**	**SPAIN MOH**	**AOGOI/SIGO/Italian MOH**	**ACOG/CDC/SMFM**	**FIGO**	**WHO**
POSTPARTUM CARE								
**Mother/child separation**	Healthy babies stay with their mothers	Separation for severely/critically ill mothers	Infants isolated and monitored for 14 days	Separation based on maternal test and symptoms	Separation based on maternal test and symptoms	Separation based on maternal test, symptoms, and willingness.	Separation for severely/critically ill mothers	Women should be supported to share a room with their baby
**Breastfeeding**	The benefits outweigh any potential risks of transmission of the virus	Can be considered for asymptomatic/mildly affected patients ****	Avoid direct breastfeeding	Allowed depending on maternal clinical state symptoms	Allowed **** /***	Pumping recommended for SARS-CoV-2 positive/PUI women ****	Can be considered for asymptomatic/mildly affected patients ****	Women can breastfeed safely **/****
	**RCOG**	**ISUOG**	**CHINA**	**SPAIN MOH**	**AOGOI/SIGO/Italian MOH**	**ACOG/CDC/SMFM**	**FIGO**	**WHO**
ANTENATAL CARE								
**Mask Wearing**	Pregnant women screened positive for SARS-CoV-2 infection	Apply local guidance	Apply local guidance	Women with acute respiratory symptoms	Pregnant women screened positive for SARS-CoV-2 infection	Women who test positive/PUI women at all times as clinically able *	People visiting a hospital or other high-risk area	NR
**Personal/social Hygiene**	Especially > 28 weeks of gestation, distance of at least two meters between individuals	Distance of at least one meter between individuals	Reduce social interactions	Reduce social interactions	Distance of at least one meter between individuals	Reduce social interactions	Distance of at least two meters or six feet between individuals	Reduce social interactions
**Antenatal visits**	Delay visits until the end of self-confinement period	Postpone routine follow-up appointments by 14 days or until positive/two consecutive negative test results	NR	NR	When possible postpone visits by taking note into the clinical file.	NR	Offer all visits for obstetric emergencies **/postpone routine follow-up appointments by 14 days or until positive/two consecutive negative test results	Guarantee to all pregnant women, including those with confirmed/suspected COVID-19, high quality of care before, during and after childbirth
**Partner/companion**	Come alone to the visits or with one person maximum	Consider reducing the number of visitors to the department	Consider reducing the number of visitors to the department.	Come alone to the visits or with one person maximum	Positive partner: notify your obstetric team: access forbidden	Consider reducing the number of visitors to the department ***	Come alone to the visits	NR
	**RCOG**	**ISUOG**	**CHINA**	**SPAIN MOH**	**AOGOI/SIGO/Italian MOH**	**ACOG/CDC/SMFM**	**FIGO**	**WHO**
INTRAPARTUM CARE								
**Antenatal Corticosteroids**	- Steroids for standard care - Do not delay urgent intervention to allow steroid administration	- Avoid steroids in critical patients - Discuss steroids administration with MDC - Avoid tocolysis	Steroids to all viable premature fetuses	- Steroids administration on individual basis - Discuss steroids administration with MTD	NR	- Steroids administration on individual basis - Particular caution for critically ill women in ICU setting	- Avoid steroids in critical patients - Discuss steroids administration with MDC - Avoid tocolysis	NR
**Respiratory Analgesia**	Entonox allowed (with disposable filter)	NR	NR	Respiratory analgesia is not recommended	NR	Consider suspending use of nitrous oxide	NR	NR
**Cord Clamping**	Delayed cord clamping recommended	Prompt cord clamping	Delayed cord clamping not recommended for women infected with COVID-19	Delayed clamping allowed if maternal and newborn isolation can be done properly	Prompt cord clamping	NR	Prompt cord clamping	NR
**Skin to skin practice**	NR	NR	NR	Skin to skin is allowed if maternal and newborn isolation can be done properly	Avoid skin to skin	NR	NR	Skin to skin is allowed
	**RCOG**	**ISUOG**	**CHINA**	**SPAIN MOH**	**AOGOI/SIGO/Italian MOH**	**ACOG/CDC/SMFM**	**FIGO**	**WHO**
POSTPARTUM CARE								
**Mother/child separation**	Healthy babies stay with their mothers	Separation for severely/critically ill mothers	Infants isolated and monitored for 14 days	Separation based on maternal test and symptoms	Separation based on maternal test and symptoms	Separation based on maternal test, symptoms, and willingness	Separation for severely/critically ill mothers	Women should be supported to share a room with their baby
**Breastfeeding**	The benefits outweigh any potential risks of transmission of the virus	Can be considered for asymptomatic/mildly affected patients ****	Avoid direct breastfeeding	Allowed depending on maternal clinical state symptoms	Allowed ****/*****	Pumping recommended for SARS-CoV-2 positive/PUI women ****	Can be considered for asymptomatic/mildly affected patients ****	Women can breastfeed safely ****/******

NR = No Recommendation provided; *: All visitors, if available, in areas of high community prevalence; **: in agreement with current local guidelines; ***: Exceptions could be made for settings of bereavement/If visitors are permitted, they should be screened for symptoms of respiratory illness Before entering a healthcare facility); ****: Mothers should ensure to wash their hands and to wear a three-ply surgical mask before touching the baby; *****: Use donated human milk when possible, especially if mother and newborn are separated; ******: keep all surfaces clean.

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
