# Peer review of "Guidelines for Pregnancy Management During the COVID-19 Pandemic: A Public Health Conundrum"

_ijerph, 2020, doi:10.3390/ijerph17218277_

Round 1
Reviewer 1 Report
Dear Authors,
The presented study tackles a very importatnt issue of Guidelines for pregnancy management during the COVID-19 pandemic: a public healthconundrum. The study was conducted reliably with appropriate selection of guidelines presented at a glance. Overall, I think that this article should be published.
However, one issues require complementary information: I suggest include informations about the type of delivery in woman with COVID-19 tests according to all guidelines.
Author Response
Dear reviewer 1, thank you for your appreciation about our manuscript and this interesting comment. Our answer is reported in blue. We hope that our modifications will improve the manuscript.
Reviewer 1:
Dear Authors,
The presented study tackles a very important issue of Guidelines for pregnancy management during the COVID-19 pandemic: a public health conundrum. The study was conducted reliably with appropriate selection of guidelines presented at a glance. Overall, I think that this article should be published. However, one issues require complementary information: I suggest include informations about the type of delivery in woman with COVID-19 tests according to all guidelines.
- We added a sentence in the introduction, Lines 49-52: “Even for delivery timing and type (vaginal or caesarean) there is a lack of evidence; therefore, decisions about route of delivery and delivery timing should be individualized for the specific patient based on obstetrical indications and maternal–foetal status.”

Reviewer 2 Report
In this manuscript, the authors performed an analysis of obstetrics guidelines, published from December 2019 to April 2020, of COVID-19 pandemic for pregnant women. They attempted to compare their recommendations and the use of maternal health care. This topic is clinically interesting for obstetricians; however, the limited knowledge about the impact of COVID-19 on pregnancy, the scientific basis of those management guidelines is relatively lacking. Consequently, the difference among those guidelines is obvious, and difficult to judge the efficacy of such guidelines. Furthermore, as the author mentioned in the introduction, neither vaccine nor effective therapies are established and widely used, the guidelines at the present time are focused on prevention and disease containment measures.
The strength of this manuscript is to assess the 11 guidelines on the obstetrics management by assessing four domains: 1) timeliness; 2) accessibility; 3) completeness; 4) consistency. The authors found only 5/11 guidelines covered all 30 foundational topics.
The weakness of this manuscript is that the authors did not perform a systematic review nor did they evaluate the impact of the different guidelines on healthcare providers or patients. Even though the present research revealed a lack of uniformity and consistency, the authors did not propose scientific answers for the best management of pregnancy, delivery, and postnatal/postpartum care. The authors just emphasized that “general recommendations should be released timely by one international body designated to lead healthcare assistance during a pandemic.” From this point of view, I am not sure whether this specific manuscript is qualified as an “original article” or “communication/research letter” for this journal.
Specific point:
In the text, the authors described that only 5/11 guidelines covered all 30 foundational topics. However, in figure 1, none of the guidelines covered all 30 topics. Please clarify this point.
Author Response
Dear reviewer 2, thank you for those interesting comments. Our answers are reported in blue. We hope that our modifications will improve the manuscript.
Reviewer 2:
In this manuscript, the authors performed an analysis of obstetrics guidelines, published from December 2019 to April 2020, of COVID-19 pandemic for pregnant women. They attempted to compare their recommendations and the use of maternal health care. This topic is clinically interesting for obstetricians; however, the limited knowledge about the impact of COVID-19 on pregnancy, the scientific basis of those management guidelines is relatively lacking. Consequently, the difference among those guidelines is obvious, and difficult to judge the efficacy of such guidelines. Furthermore, as the author mentioned in the introduction, neither vaccine nor effective therapies are established and widely used, the guidelines at the present time are focused on prevention and disease containment measures.
The strength of this manuscript is to assess the 11 guidelines on the obstetrics management by assessing four domains: 1) timeliness; 2) accessibility; 3) completeness; 4) consistency. The authors found only 5/11 guidelines covered all 30 foundational topics.
The weakness of this manuscript is that the authors did not perform a systematic review nor did they evaluate the impact of the different guidelines on healthcare providers or patients. Even though the present research revealed a lack of uniformity and consistency, the authors did not propose scientific answers for the best management of pregnancy, delivery, and postnatal/postpartum care. The authors just emphasized that “general recommendations should be released timely by one international body designated to lead healthcare assistance during a pandemic.” From this point of view, I am not sure whether this specific manuscript is qualified as an “original article” or “communication/research letter” for this journal.
Specific point:
In the text, the authors described that only 5/11 guidelines covered all 30 foundational topics. However, in figure 1, none of the guidelines covered all 30 topics. Please clarify this point.
- This is right; we added in the text line 22 of the abstract and line 178-183 of the main text the explanation of the number of guidelines that covered more than 80% of the 30 foundational topics.
- We also added a paragraph about limitations of this manuscript. We added a paragraph at line 332-334: “The main limitations of this analysis are that we did not perform a systematic review and we based the assessment of the guidelines’ completeness by establishing 30 foundational topics based on our clinical experience. In addition, we did not evaluate the impact of the different guidelines on healthcare providers or patient”

Reviewer 3 Report
Title: Guidelines for pregnancy management during the COVID-19 pandemic: a public health conundrum
The objective of this preliminary expert review was to examine the 11 obstetrics guidelines released from December 2019 to April 2020 and compare their recommendations and assess how useful they could be to healthcare workers. Overall, this study provides an interesting description comparing the major recommendations on obstetrics during the COVID-19 pandemic. However, there are some major and minor concerns outlined below.
- In the abstract, I would remove the first sentence “Pregnant women are at high risk for developing complications of COVID-19.” because at this time, I don’t think we are 100% sure with sufficient evidence to boldly state this.
- Accessibility (lines 110-115): In addition to the technical accessibility (i.e. number of clicks), was language considered? To be accessible globally, I think the guidelines need to be in English format. But for implementation to the local healthcare providers, a translated version may be necessary.
- In Table 1, could authors clarify what the different publication types mean? Were they all uploaded on a website? It would be helpful if this table had 1) in what format the recommendation was released (e.g. paper, website, email, etc, which will give an overview of how people can access to these information) and how they were formatted (e.g. commentary, guidelines, interim guidance, communication/Q&A, etc, indicating if they were systematically structured guidelines or not). Currently, I think there is a mix of these two under “publication type”.
Also, for publication dates and last updated dates, please add (date/months) under the heading so it’s clear what these number in fractions mean (it was not obvious to me).
Since the authors report the number of updates in the text, I would suggest adding a column for number of updated versions (e.g. n=7 for RCOG).
It would be helpful to have a column of the country of the institution or have the country in parentheses after the name of the releasing institutions.
For the column on “30 Foundational Topics covered (there is a typo in the column heading)”, I would like to see the number out of the 30 topics that was covered instead of having YES/NO which will help us get a more precise understanding of how much of the topics were covered in each recommendation. Also, the selection of 80% as cutoff for YES/NO seems misleading.
I am not sure if the current rows in Table 1 are ordered in a specific way but I would suggest reordering the recommendation list by the first release date (so RCOG comes first) since I would assume that time will influence how detailed and fleshed the recommendations are (e.g. number of Fundamental topics covered) and whether they were updated or not.
- Please clarify this sentence in line 162-163 “The completeness of the publications depended on the type of release (with ‘guidelines’ in pdf 162 format being the most complete and the ‘question-and-answer’ format the least detailed) and on the 163 number of references cited (0 for WHO, 79 for FIGO).” Do the authors mean that recommendations published in guideline format and having greater number of references were more likely to be “complete” measured via number of foundational topics covered? I think the term “publications depended on” may be misleading.
- Figure 2 title has a typo: “divided”
- Instead of Figure 1, I would like to see a table with the 30 fundamental topics and see exactly which recommendations covered which fundamental topic (with tick marks). This descriptive table will help the reader understand section 3.3. more in detail. Figure 1 and Figure 2 is overall showing the same results.
- Table 2 looks great but would like the authors to modify for clarity. Overall, the blurbs in the cell are not self-explanatory and needs to be consistent. For example, in the “Mask Wearing” row, RCOG and AOGOI indicates the targeted women for mask wearing (just subject) but the other cells have sentences. I would suggest to revise this table so the blurbs in the cell are all sentences or phrases to be consistent. Also, please make sure to be consistent on whether to end with periods or not. Some end with periods but some don’t.
- Lines 251-254 is stating the observation in Table 1 so should be under the result section.
- In the discussion section, the authors only discuss about the completeness of the recommendations. I would like the authors to discuss about timeliness, accessibility, and consistency of the recommendations as well. Specifically including how the timeliness may have influenced the completeness of the recommendations and how accessibility may impact management.
- I think the authors should also discuss any limitations about measuring completeness using the 30 foundational topics the authors defined. Also, while authors highlight the existence of 10 points of conflict out of the 30, can authors discuss about the missingness of the topics across the recommendations as well (were there any patterns)? Again, it would help to have a grid showing which recommendation addressed which topics.
- In general, guidelines are created based on reliable evidence. However, while there was an urgent need for guidance, I would imagine all institutions having difficulties presenting obstetrics guidelines while the COVID-19 pandemic emerged so rapidly. In this context, I would like the authors to discuss the value of timeliness but also updating their recommendations as new evidence emerges.
- Authors state in the discussion that the differences in the recommendations may be reflective of the different healthcare systems. I agree with this sentence but could the authors elaborate more on this with specific examples? Also, it seems most of the recommendations reviewed in the manuscript are from developed countries. Could authors also discuss about how the timeliness, accessibility, completeness, and consistency of these recommendations may have influenced the obstetric practice in developing countries?
- I do not agree that this manuscript supports the conclusion of “General recommendations should be released in a timely manner by one international body designated to lead healthcare assistance during a pandemic.” I agree with timely manner but the authors don’t discuss the pros and cons about one single international body giving out recommendations. This needs to be discussed and justified in more detail if this was to be stated in the conclusion.
While the authors say they have no intentions to propose scientific answers for generating management, the conclusion does sound like they are proposing that one international body should be designated to lead guidance during a pandemic. I suggest the authors to at minimum, take out the last sentence from the abstract and tone this down in the discussion and conclusion of the manuscript.
Author Response
Dear reviewer, thank you for taking the time to analyze our manuscript and for those interesting comments. Our answers are reported in blue. We hope that our modifications (and new table) will improve the manuscript.
Reviewer 3:
The objective of this preliminary expert review was to examine the 11 obstetrics guidelines released from December 2019 to April 2020 and compare their recommendations and assess how useful they could be to healthcare workers. Overall, this study provides an interesting description comparing the major recommendations on obstetrics during the COVID-19 pandemic. However, there are some major and minor concerns outlined below.
- In the abstract, I would remove the first sentence “Pregnant women are at high risk for developing complications of COVID-19.” because at this time, I don’t think we are 100% sure with sufficient evidence to boldly state this.
- This is correct. We modified the first sentence in the abstract line 13 according to the last evidence but if you prefer we can delete it.
- Accessibility (lines 110-115): In addition to the technical accessibility (i.e. number of clicks), was language considered? To be accessible globally, I think the guidelines need to be in English format. But for implementation to the local healthcare providers, a translated version may be necessary.
- Line 119 We added a sentence to complete our definition of accessibility including language. Line 171-173 We added a sentence about accessibility considering languages.
- In Table 1, could authors clarify what the different publication types mean? Were they all uploaded on a website? It would be helpful if this table had 1) in what format the recommendation was released (e.g. paper, website, email, etc, which will give an overview of how people can access to these information) and how they were formatted (e.g. commentary, guidelines, interim guidance, communication/Q&A, etc, indicating if they were systematically structured guidelines or not). Currently, I think there is a mix of these two under “publication type”.
Also, for publication dates and last updated dates, please add (date/months) under the heading so it’s clear what these number in fractions mean (it was not obvious to me).
Since the authors report the number of updates in the text, I would suggest adding a column for number of updated versions (e.g. n=7 for RCOG).
It would be helpful to have a column of the country of the institution or have the country in parentheses after the name of the releasing institutions.
For the column on “30 Foundational Topics covered (there is a typo in the column heading)”, I would like to see the number out of the 30 topics that was covered instead of having YES/NO which will help us get a more precise understanding of how much of the topics were covered in each recommendation. Also, the selection of 80% as cutoff for YES/NO seems misleading.
I am not sure if the current rows in Table 1 are ordered in a specific way but I would suggest reordering the recommendation list by the first release date (so RCOG comes first) since I would assume that time will influence how detailed and fleshed the recommendations are (e.g. number of Fundamental topics covered) and whether they were updated or not.
- Modifications have been done according to your comments, please see the new Table 1.
- Please clarify this sentence in line 162-163 “The completeness of the publications depended on the type of release (with ‘guidelines’ in pdf 162 format being the most complete and the ‘question-and-answer’ format the least detailed) and on the 163 number of references cited (0 for WHO, 79 for FIGO).” Do the authors mean that recommendations published in guideline format and having greater number of references were more likely to be “complete” measured via number of foundational topics covered? I think the term “publications depended on” may be misleading.
- We added a sentence line 182-183 to explain that we observed that papers written in form of guideline and those with more references were also the most complete ones.
- Figure 2 title has a typo: “divided”
- Corrected, thank you.
- Instead of Figure 1, I would like to see a table with the 30 fundamental topics and see exactly which recommendations covered which fundamental topic (with tick marks). This descriptive table will help the reader understand section 3.3. more in detail. Figure 1 and Figure 2 is overall showing the same results.
- We have added a new Table 2 named “Foundational topics covered by each guideline”. Ex Table 2 “Main differences among the 11 guidelines” became Table 3.
- Table 2 looks great but would like the authors to modify for clarity. Overall, the blurbs in the cell are not self-explanatory and needs to be consistent. For example, in the “Mask Wearing” row, RCOG and AOGOI indicates the targeted women for mask wearing (just subject) but the other cells have sentences. I would suggest to revise this table so the blurbs in the cell are all sentences or phrases to be consistent. Also, please make sure to be consistent on whether to end with periods or not. Some end with periods but some don’t.
- Please see modifications according to your comment in Table 3 (ex Table 2).
- Lines 251-254 is stating the observation in Table 1 so should be under the result section.
- We reviewed this sentence and put it under the result section lines 154-157.
- In the discussion section, the authors only discuss about the completeness of the recommendations. I would like the authors to discuss about timeliness, accessibility, and consistency of the recommendations as well. Specifically including how the timeliness may have influenced the completeness of the recommendations and how accessibility may impact management.
- We added a new paragraph at lines 284-291: “Even if the first guidelines released may have been not as complete as those following, timeliness did not interfere with completeness, especially when the guidelines were frequently updated. The best example from the analysis is represented by the RCOG guidelines, released first and updated 7 times in the observed period. Releasing timely information is indeed fundamental to guide HCP: further details can then be provided once more data are available. Accessibility is another feature of paramount importance for guidelines. To best support health care professionals in managing their patients in such an evolving scenario, guidelines should be easily accessible from a website/mailing list, and written both in English and in the local language.
- I think the authors should also discuss any limitations about measuring completeness using the 30 foundational topics the authors defined. Also, while authors highlight the existence of 10 points of conflict out of the 30, can authors discuss about the missingness of the topics across the recommendations as well (were there any patterns)? Again, it would help to have a grid showing which recommendation addressed which topics.
- We added a paragraph at line 332-334: “The main limitations of this analysis are that we did not perform a systematic review and we based the assessment of the guidelines’ completeness by establishing 30 foundational topics based on our clinical experience. In addition, we did not evaluate the impact of the different guidelines on healthcare providers or patient”
- We completed a paragraph at lines 207-211: “As highlighted in Table 2, no guideline gives indications about family planning, and the majority did not address the management of biohazardous materials neither the use of Magnesium Sulphate and anticoagulant therapies and the mode of transport of the patients from home to the health care centers. We also observed that only 4 guidelines discuss the problem of mental health, as well as the management of labour and foetal monitoring.”
- We have added a new table, which became Table 2: “Foundational topics covered by each guideline”.
- In general, guidelines are created based on reliable evidence. However, while there was an urgent need for guidance, I would imagine all institutions having difficulties presenting obstetrics guidelines while the COVID-19 pandemic emerged so rapidly. In this context, I would like the authors to discuss the value of timeliness but also updating their recommendations as new evidence emerges.
- We responded to this comment, together with comment number 9, in lines 284-291.
- Authors state in the discussion that the differences in the recommendations may be reflective of the different healthcare systems. I agree with this sentence but could the authors elaborate more on this with specific examples? Also, it seems most of the recommendations reviewed in the manuscript are from developed countries. Could authors also discuss about how the timeliness, accessibility, completeness, and consistency of these recommendations may have influenced the obstetric practice in developing countries?
- We added a paragraph to answer this interesting comment line 309-314: “It was a major challenge for developing countries to access, assess and define the right recommendations for obstetric care. In fact, several aspects of some guidelines on maternal and newborn health were not applicable in low-resource countries. One of the authors (CB), for example, worked with the Ministry of Health of Madagascar to review and adapt obstetric guidelines to their specific context. “
- I do not agree that this manuscript supports the conclusion of “General recommendations should be released in a timely manner by one international body designated to lead healthcare assistance during a pandemic.” I agree with timely manner but the authors don’t discuss the pros and cons about one single international body giving out recommendations. This needs to be discussed and justified in more detail if this was to be stated in the conclusion.
- In response, we have added the following sentence at lines 320-324: “This could be accomplished through the harmonizing role of a leading international body that could define a template to be used to generate guidelines during a pandemic, including the foundational topics, and could receive and release the guidelines from different authorities, in order to update continuously at a central level the shared guidance available worldwide for managing a pandemic.”
- While the authors say they have no intentions to propose scientific answers for generating management, the conclusion does sound like they are proposing that one international body should be designated to lead guidance during a pandemic. I suggest the authors to at minimum, take out the last sentence from the abstract and tone this down in the discussion and conclusion of the manuscript.
- Lines 349-352: we deleted the last sentence, as suggested: General recommendations should be released in a timely manner by one international body designated to lead healthcare assistance during a pandemic. Such easy-to-access recommendations could then be adopted and adapted locally, depending on the context and resources available in each country.

Round 2
Reviewer 3 Report
The authors have addressed all of the concerns.